# Korean Validation of the Short Version of the TEMPS-A (Temperament Evaluation of Memphis, Pisa, Paris, and San Diego Autoquestionnaire) in Patients with Mood Disorders

**DOI:** 10.3390/medicina59010115

**Published:** 2023-01-06

**Authors:** Sunho Choi, Hyeona Yu, Joohyun Yoon, Yoonjeong Jang, Daseul Lee, Yun Seong Park, Hong Kyu Ihm, Hyun A Ryoo, Nayoung Cho, Jong-Min Woo, Hyo Shin Kang, Tae Hyon Ha, Woojae Myung

**Affiliations:** 1Department of Psychiatry, Seoul National University College of Medicine, Seoul 03080, Republic of Korea; 2Department of Neuropsychiatry, Seoul National University Bundang Hospital, Seongnam 13619, Republic of Korea; 3Seoul Mental Health Clinic, Seoul 06149, Republic of Korea; 4Department of Psychology, Kyungpook National University, Daegu 41566, Republic of Korea

**Keywords:** temperament, measurement, mood disorder, validation, Korean

## Abstract

Background and Objectives: The Temperament Evaluation of Memphis, Pisa, Paris and San Diego Autoquestionnaire (TEMPS-A) is designed to assess affective temperaments. The short version of the TEMPS-A (TEMPS-A-SV) has been translated into various languages for use in research and clinical settings. However, no research has been conducted to validate the Korean version of the TEMPS-A-SV in patients with mood disorders. The goal of this study is to evaluate the reliability and validity of the TEMPS-A-SV in Korean mood disorder patients. Materials and Methods: In this cross-sectional retrospective study, a total of 715 patients (267 patients with major depressive disorder, 94 patients with bipolar disorder I, and 354 patients with bipolar disorder II) completed the Korean TEMPS-A-SV. Cronbach’s alpha and McDonald’s omega were used to assess the reliability. Exploratory factor analysis (EFA) was also performed. Spearman’s correlation coefficient was used to examine associations between the five temperaments. The difference in five temperament scores between the gender or diagnosis groups was analyzed, and the correlation between five temperament scores and age was tested. Results: The Korean TEMPS-A-SV displayed good internal consistency (α = 0.65–0.88, ω = 0.66–0.9) and significant correlations between the subscales except one (the correlation between hyperthymic and anxious). Using EFA, a two-factor structure was produced: Factor I (cyclothymic, depressive, irritable, and anxious) and Factor II (hyperthymic). The cyclothymic temperament score differed by gender and the anxious temperament score was significantly correlated with age. All the temperaments, except for irritable temperament, showed significant differences between diagnosis groups. Conclusions: Overall, the results show that the TEMPS-A-SV is a reliable and valid measurement that can be used for estimating Koreans’ affective temperaments. However, more research is required on affective temperaments and associated characteristics in people with mood disorders.

## 1. Introduction

In 5th century B.C., Greek physician Hippocrates suggested that the four humors (blood, phlegm, yellow bile, and black bile) significantly contribute to physical and psychological health [1]. Regarding psychological health, the four humors are considered to be associated with temperaments: blood with the sanguine type, phlegm with the phlegmatic type, yellow bile with the choleric type, and black bile with the melancholic type. This idea later led Kraepelin to introduce four basic affective temperaments: cyclothymia, depression, mania, and irritability [2], and Akiskal et al. later added anxious temperament to the concept and made a five-temperament system [3].

The Temperament Evaluation of Memphis, Pisa, Paris, and San Diego Autoquestionnaire (TEMPS-A) was developed by Akiskal, et al. [4] to evaluate five affective temperaments. It is a self-report questionnaire comprising 110 items. The TEMPS-A has effectively compared patients with mood disorders and between individuals with or without relatives who are patients [5,6] and made predictions of medication adherence and clinical course in patients [7,8,9]. The TEMPS-A has been translated into different language versions including Polish [10], German [11], Italian [12], Japanese [13], Arabic [14], and Korean [15], for further use. However, it is difficult to apply the full-length TEMPS-A, which comprises 110 items, for patients with severe symptoms in the clinical setting because it demands a lot of time and effort. Akiskal and colleagues created a short 39-item version of the TEMPS-A to make evaluation easier, thus requiring less time and effort. This short version of the TEMPS-A has been translated into diverse languages including German [11], Italian [16], Chinese [17], and Spanish [18]. The TEMPS-A-SV also has significant efficacy and usefulness as it can be utilized to distinguish between patients with major depression disorders and patients with bipolar disorders [17,19]. Hyperthymic, cyclothymic, and anxious temperament scores are found to be able to differentiate patients with BDs from those with MDDs. Moreover, the TEMPS-A-SV has proven that it can be used as a screening tool to detect non-clinical individuals who are susceptible to mood disorder [17].

The TEMPS-A-SV has not yet been developed and validated in Korean, despite its significance. Therefore, the goal of this study is to demonstrate the validity of the Korean TEMPS-A (short version) in a clinical population. Additionally, we aim to compare the temperaments of patients with different mood disorders.

## 2. Materials and Methods

### 2.1. Subjects

This study analyzed data from 715 patients (267 patients with major depressive disorder [MDD], 94 patients with bipolar disorder I [BD I], and 354 patients with bipolar disorder II [BD II]) from July 2013 to February 2021. The sample comprised 222 men and 493 women ranging from 16 to 71 years old. The patients were from the mood disorder clinic of Seoul National University Bundang Hospital (SNUBH). Their medical records were used to acquire all pertinent information. Based on a structured diagnostic interview (Mini-International Neuropsychiatric Interview [M.I.N.I]) [20] and review of case files, two board-certified psychiatrists (T.H.H. and W.M.) confirmed the patients’ diagnosis in accordance with the Diagnostic and Statistical Manual of Mental Disorders, Fifth Edition (DSM-5) [21]. We also gathered additional pertinent data, such as age, gender, education level, employment status, marital status, alcohol use, smoking history, and history of psychiatric hospitalizations. The Institutional Review Board of Seoul National University Bundang Hospital gave its approval for this study (protocol code B-2205-756-111, approved 2 May 2022). Because information was gathered retroactively through a review of medical records, patient informed consent was waived. 

### 2.2. Measurement and Procedures

Five temperamental features (cyclothymic, depressive, irritable, hyperthymic, and anxious) were evaluated using the short version of the Korean TEMPS-A questionnaire (39 questions). The following were the items that belonged to each subscale: cyclothymic temperament comprised items 1–12 (12 questions), depressive temperament comprised items 13–20 (8 questions), irritable temperament comprised items 21–28 (8 questions), hyperthymic temperament comprised items 29–36 (8 questions), and anxious temperament comprised items 37–39 (3 questions). The TEMPS-A-SV is a self-report scale, and each item received a dichotomous score (Yes = 1 and No = 0).

Two psychiatrists (THH and WM) translated the TEMPS-A-SV into Korean and reviewed the items. The translated version was further forwarded to psychiatrists, psychologists, psychiatric nurses, and clinical researchers for their expert opinion. After the review, the questionnaire was trimmed appropriately to improve legibility and fluency.

### 2.3. Statistical Analysis

This study used mean values as the scores for each temperament. Mean values were calculated as follows: the sum of the items’ scores of each subscale/the number of items. The Shapiro–Wilk test was used to ascertain if the score distribution was normal. Non-parametric analyses were used since the parameters did not follow a normal distribution. Spearman’s correlation test was employed to test the correlation between five temperaments. Spearman’s partial correlation test was used to test the correlation between five temperaments and age after adjusting for covariates (gender, primary diagnosis, education level, employment status, marital status, first-degree family history of psychiatric disorders, alcohol use, and smoking history). Median regression was conducted to ascertain differences in scores of five temperaments according to gender or diagnoses (MDD, BD I, or BD II) after controlling covariates. For comparison, the temperament scores were presented as the median and interquartile range. Cronbach’s alpha and McDonald’s omega were used to evaluate the internal consistency of the questionnaire and its subscales. Exploratory factor analysis was performed with varimax rotation. R 4.1.2 (Vienna, Austria; http://R-project.org/, accessed on 1 November 2021) software and Stata SE 17 (https://www.stata.com/, accessed on 20 April 2021) were used for analyses.

## 3. Results

### 3.1. Clinical and Demographic Characteristics

The participants were 34.5 years old on average (SD = 12.4), with 69% of them being female. Table 1 shows the other clinical and demographic characteristics.

### 3.2. Reliability Analysis

Table 2 shows the Cronbach’s alpha coefficients and McDonald’s omega coefficients of individual temperaments (cyclothymic, depressive, irritable, hyperthymic, and anxious) and the whole questionnaire. Cronbach’s alpha values for the five temperaments were 0.85 (cyclothymic), 0.78 (depressive), 0.79 (irritable), 0.74 (hyperthymic), and 0.65 (anxious). Cronbach’s alpha value for the whole questionnaire was 0.88. McDonald’s omega coefficients for the five temperaments were 0.87 (cyclothymic), 0.83 (depressive), 0.82 (irritable), 0.79 (hyperthymic), and 0.66 (anxious). McDonald’s omega coefficient for the whole questionnaire was 0.9. All alpha and omega coefficients were higher than 0.6.

### 3.3. Exploratory Factor Analysis

The results of the EFA performed using varimax rotation are shown in Table 3. By integrating factors with loadings higher than 0.4, two superfactors were generated. Cyclothymic, depressive, irritable, and anxious temperaments were contained in Factor I (33.0%), but only the hyperthymic temperament was present in Factor II (17.6%). The projection of the results of the factor analysis is shown in Figure 1. The path diagram for the factor analysis is further displayed in Appendix A.

### 3.4. Correlations within the Scales of the TEMPS-A-SV

Table 4 displays the outcomes of the subscales’ Spearman’s correlation test. All the correlations among the temperaments except one (the correlation between hyperthymic and anxious) were significant. Furthermore, only the depressive and hyperthymic temperaments were negatively correlated (*rho* = −0.12, *p* < 0.01). Cyclothymic and irritable temperaments displayed the highest positive correlation of all correlations (*rho* = 0.50, *p* < 0.001). Other correlations among the temperaments were as follows: cyclothymic and depressive temperaments (*rho* = 0.49, *p* < 0.001), depressive and irritable temperaments (*rho* = 0.44, *p* < 0.001), depressive and anxious temperaments (*rho* = 0.32, *p* < 0.001), cyclothymic and anxious temperaments (*rho* = 0.31, *p* < 0.001), irritable and hyperthymic temperaments (*rho* = 0.25, *p* < 0.001), irritable and anxious temperaments (*rho* = 0.23, *p* < 0.001), and cyclothymic and hyperthymic temperaments (*rho* = 0.18, *p* < 0.001).

### 3.5. Correlations between Five Temperaments and Age

Table 5 displays the findings of the partial Spearman’s correlation test between five temperaments and age. All temperaments, except anxious temperament, were significantly and negatively correlated with age: depressive (*rho* = −0.27, *p* < 0.001), cyclothymic (*rho* = −0.26, *p* < 0.001), irritable (*rho* = −0.15, *p* < 0.001), and hyperthymic (*rho* = −0.08, *p* < 0.05). 

### 3.6. Discrepancies between Groups in TEMPS-A-SV Scores

The median regression analysis showed that the TEMPS-A-SV scores between the two genders and three patient groups (MDD, BD I, and BD II) differed significantly (Figure 2). 

Females scored much higher on the cyclothymic scale than males did (female: [0.583, 0.333–0.833] vs. male: [0.500, 0.25–0.75], *p* < 0.01). The score of cyclothymic temperament of the BD II group (0.667, 0.417–0.833) was significantly higher than that of the MDD (0.417, 0.167–0.667, *p* < 0.001) and the BD I groups (0.458, 0.167–0.667, *p* < 0.001). Regarding depressive temperaments, the score of the BD II group (0.500, 0.25–0.75) was significantly higher than that of the BD I group (0.250, 0.125–0.594, *p* < 0.001). Regarding hyperthymic temperaments, the score of the BD I group (0.375, 0.125–0.750) was significantly higher than that of the MDD group (0.250, 0.125–0.500, *p* < 0.01). Regarding anxious temperaments, the score of the BD I group (0.333, 0–0.667) was significantly lower than that of the MDD (0.667, 0–0.667, *p* < 0.001) and the BD II groups (0.667, 0–1.000, *p* < 0.05).

## 4. Discussion

This study developed and validated the short version of the Korean TEMPS-A. The Korean TEMPS-A-SV demonstrated strong reliability and validity. Regarding reliability, Cronbach’s alpha and McDonald’s omega coefficients indicated significant internal consistency. All coefficients, except anxious temperament, were higher than the commonly accepted standard of 0.7: cyclothymic (α = 0.85, ω = 0.87), depressive (α = 0.78, ω = 0.83), irritable (α = 0.79, ω = 0.82), hyperthymic (α = 0.74, ω = 0.79), and anxious (α = 0.65, ω = 0.66). Similar to our study, some TEMPS-A-SV studies of other languages have also reported coefficients that are lower than 0.7 [22,23,24,25]. In addition, as Cronbach’s alpha and McDonald’s omega coefficients are influenced by the number of items within a scale [26,27], α = 0.65 and ω = 0.66 were considered acceptable values, considering the small number of items evaluating anxious temperament (three items). Nevertheless, the reliability coefficients of anxious temperament subscale implies that there is a need to refine this subscale to strengthen its internal consistency.

Our results on the validity of the TEMPS-A are similar to those of previous studies for other languages [28,29,30,31]. By exploratory factor analysis, five temperaments were divided into two superfactors, i.e., one with cyclothymic (0.69), depressive (0.76), irritable (0.65), and anxious (0.42) temperaments, and the other one with a hyperthymic (0.87) temperament. This result is also similar to the results of other validation studies [30,31], which also found a two-factor structure. Such structure was not only found in studies validating the TEMPS-A-SV but also in studies investigating full versions of the TEMPS-A in different languages [28,29]. The consistency in structure strongly supports the construct validity of the short version of the Korean TEMPS-A. 

Spearman’s correlation noted prominent correlations among the cyclothymic, depressive, and irritable temperaments: cyclothymic and depressive (r = 0.49), cyclothymic and irritable (r = 0.50), and depressive and irritable (r = 0.44) (Table 4). The correlations among these temperaments were consistent with the previous validation of the Italian short version in healthy controls [16]. Cyclothymic, depressive, and irritable temperaments among BD and MDD patients were also significantly correlated [32,33]. There was a tenuous negative association between hyperthymic and depressive temperaments (r = −0.12), which is consistent with earlier research [29,34]. This result was expected because hyperthymic temperament is considered a protective factor for psychiatric disorders, including depression [35].

By using Spearman’s partial correlation, we found that all correlations between age and temperaments, with the exception of the anxious temperament, were significant. All temperaments, except the anxious temperament, were negatively correlated with age: cyclothymic (r = −0.26, *p* < 0.001), depressive (r = −0.27, *p* < 0.001), irritable (r = −0.15, *p* < 0.001), and hyperthymic (r = −0.08, *p* < 0.05). Although the research on the correlation between age and temperaments is scant, this result was slightly different from previous studies. A review paper of several TEMPS-A studies in the non-clinical population noted that age only affected the depressive temperament [36]. Moreover, Kawamura, et al. [37] found that temperaments remained constant throughout time in a six-year longitudinal study. There are two speculations relating to the difference in findings. First, the difference in study population (general population vs. mood disorder patients) could affect correlation between temperaments and age. For instance, a recent TEMPS-A validation among patients with bipolar or cyclothymic disorder has noted that temperaments, with the exception of hyperthymic temperament, are negatively correlated with age [38]. Additionally, this difference in correlations may be due to our sample group’s age range. The correlations of temperaments with age tend to be more significant when the standard deviation of age in the sample group is bigger [14,39,40]. Since the standard deviation of age in our sample is quite big (SD = 12.4), covering a larger age range, our analyses were more likely to identify correlations between temperaments and age. Lastly, as culture significantly influences temperaments, cultural differences might drive the discrepancy [41]. Previous studies have noted that attitudes toward self-expression are different in Western cultures and non-Western cultures, which can offer a foundation for developing temperamental traits [42].

Our study compared TEMPS-A scores in men and women by median regression. Only cyclothymic temperament showed significant differences. Women scored considerably higher on the cyclothymic temperament scale than males, which is consistent with earlier research [36,43]. Considering that patients with BD II are more likely to have cyclothymic temperaments than those with BD I [44], our result could support findings which suggest that bipolar II disorders are more common in women than bipolar I disorders [45,46,47].

The TEMPS-A scores differed according to the diagnosis of mood disorders (Figure 2). First, the BD II group showed significantly higher scores in cyclothymic temperaments than MDD and BD I groups. Previous studies have suggested that cyclothymic temperaments may help to distinguish BD II from MDD [48,49]. The findings of our study are in line with those of several previous studies. This finding suggests that an assessment of temperaments among patients with mood disorders could contribute to differential diagnoses between BD II and MDD. Previous studies noted that, unlike BD I, the most prevalent temperament among BD II patients is cyclothymic [44], and that the cyclothymic temperament is a particularly sensitive indicator of BD II [50]. Thus, our results provide evidence for temperamental mood lability being a risk factor for developing bipolar II disorder [50,51,52]. There is a need for further research to comprehensively ascertain the cyclothymic temperament differences between MDD, BD I, and BD II.

The depressive scores between BD I and BD II groups differed significantly. The BD II group showed significantly higher scores in depressive temperaments than the BD I group. Although no study has compared the TEMPS-A scores of patients with BD I and BD II, research has shown that the proportion of time spent in depression and recurrence rates was higher in the BD II group than in the BD I group [53]. BD II patients have longer and more severe periods of depression than BD I patients, and this may lead to higher depressive scores of BD II patients.

Regarding hyperthymic temperament, significantly higher scores were obtained by the BD I group than by the MDD group. However, there was no significant difference in the hyperthymic score between the MDD and BD II groups. Previous studies reported mixed results for the difference in hyperthymic temperaments between MDD and BD II [54,55,56]. The varying results of studies may be due to the differing sample sizes. 

Compared to the MDD and BD II groups, the BD I group had considerably lower scores on the anxious temperament. Previous researchers sought to ascertain the differences in the anxious temperament between MDDs and BDs in several ways, but various results were found: no differences were found between the two groups [57], the BD group was found to be more anxious [58], and the MDD group was found to be more anxious [59]. Considering the fact that these research studies did not differentiate between the two types of BD, we can speculate that these mixed results may be due to the different level levels anxious temperament in BD I and BD II. As BD I showed lower anxious temperament scores than the MDD group and the BD II group showed similar scores to the MDD group, mixed findings can be found when BD I and BD II groups are integrated. Although further studies are essential to identify the differences in anxious temperament scores of the MDD and BD groups, our results contribute to the discussion of the anxious temperament in MDD and BD patients.

This study has several limitations. First, because only clinical participants were included in the validation, there is a need for further studies that apply the TEMPS-A-SV to examine the general Korean population. Second, this study was a cross-sectional retrospective study; thus, follow-up assessments were not conducted. Lastly, test–retest reliability was not evaluated. Despite these limitations, it is notable that our study is the first to validate the short version of the Korean TEMPS-A and suggest the clinical correlates of the TEMPS-A among patients with mood disorders.

## 5. Conclusions

In this study, the short version of the Korean TEMPS-A performed well in terms of validity and reliability among a clinical population. Cronbach’s alpha and McDonald’s omega coefficients were significant, and exploratory factor analysis found two superfactors among the five temperaments, in line with previous research. TEMPS-A score variations between patient groups implied some new findings about mood disorders, although further studies are required. Our study is the first to adopt the short version of the Korean TEMPS-A, and more studies could be conducted to support this study or to make new findings. The short version of the Korean TEMPS-A could be validated in the non-clinical sample to compare the TEMPS-A scores between the clinical and non-clinical sample. Furthermore, the cultural effects on the TEMPS-A scores could be examined based on results from diverse countries.

## Figures and Tables

**Figure 1 medicina-59-00115-f001:**
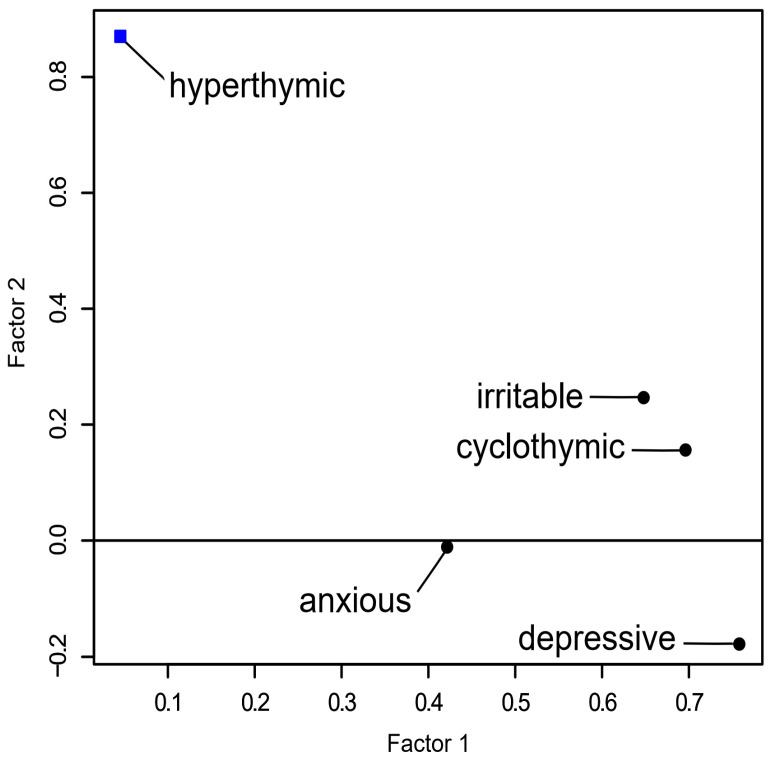
Exploratory factor analysis factor plot of five temperaments (varimax rotation).

**Figure 2 medicina-59-00115-f002:**
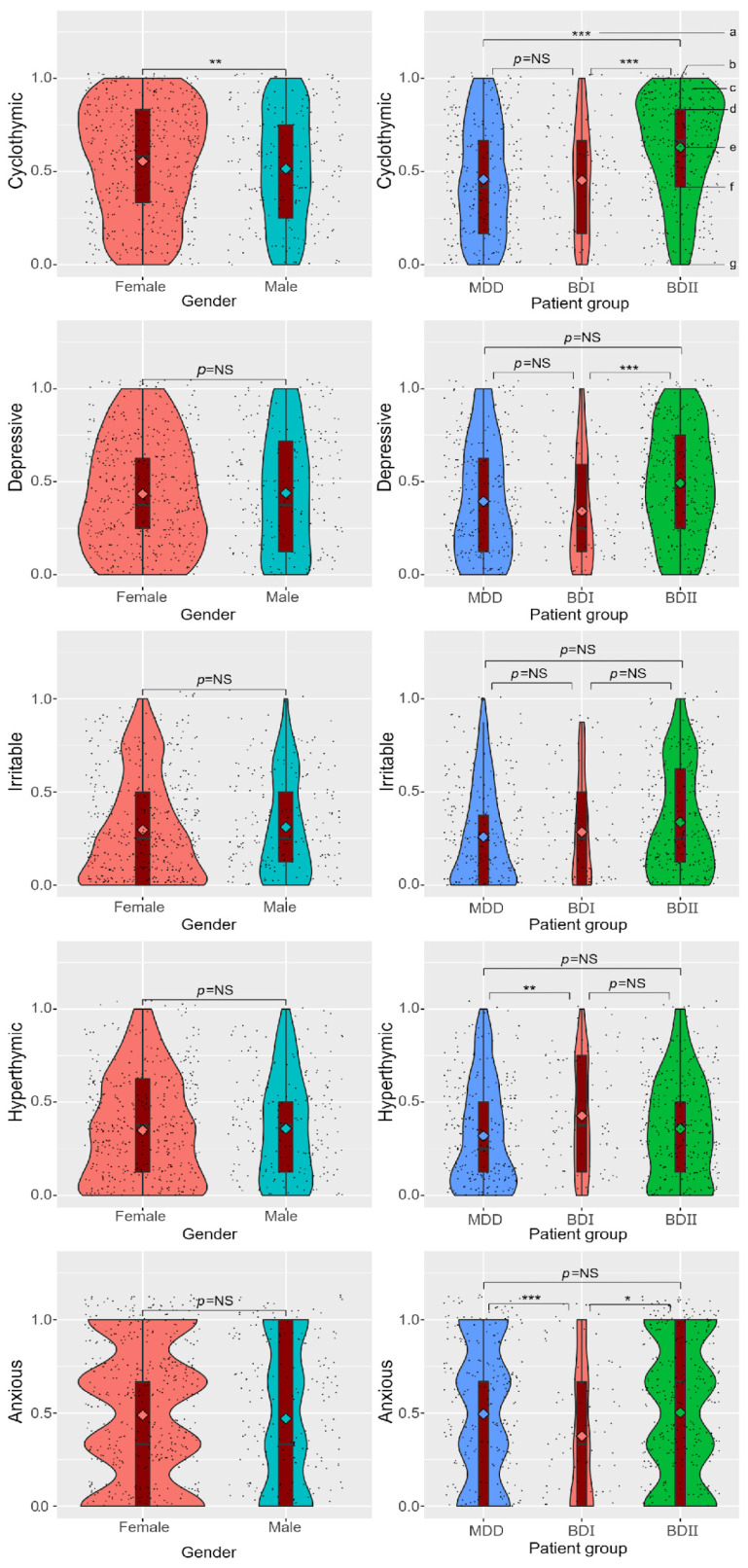
Violin plots depicting the distribution of five temperaments by gender or patient group. (a) *p*-value of median regression; * *p* < 0.05, ** *p* < 0.01, *** *p* < 0.001; NS = no significance. (b) Upper adjacent value. (c) Distribution of temperaments. (d) Third quartile (75%ile). (e) Median. (f) First quartile (25%ile). (g) Lower adjacent value.

**Table 1 medicina-59-00115-t001:** Participants’ demographic and clinical features (*n* = 715).

Characteristics	All Patients (*n* = 715)	Major Depressive Disorder Patients (*n* = 267)	Bipolar Disorder I Patients (*n* = 94)	Bipolar Disorder II Patients (*n* = 354)	*p* ^1^
Age (years)	34.5 ± 12.4	38.8 ± 12.6	33.0 ± 11.2	31.8 ± 11.6	<0.001
Gender (%)					0.043
Male	222 (31.0%)	70 (26.2%)	37 (39.4%)	115 (32.5%)	
Female	493 (69.0%)	197 (73.8%)	57 (60.6%)	239 (67.5%)	
Education (%)					0.787
High school or below	489 (68.4%)	180 (67.4%)	67 (71.3%)	242 (68.4%)	
Others	226 (31.6%)	87 (32.6%)	27 (28.7%)	112 (31.6%)	
Employment status (%)					0.616
Employed	249 (34.8%)	99 (37.1%)	32 (34.0%)	118 (33.3%)	
Unemployed	466 (65.2%)	168 (62.9%)	62 (66.0%)	236 (66.7%)	
Marital status (%)					<0.001
Married	267 (37.3%)	139 (52.1%)	25 (26.6%)	103 (29.1%)	
Others (single, divorced, or widowed)	448 (62.7%)	128 (47.9%)	69 (73.4%)	251 (70.9%)	
Psychiatric family history (%)					0.303
Yes	235 (32.9%)	80 (30.0%)	29 (30.9%)	126 (35.6%)	
No	480 (67.1%)	187 (70.0%)	65 (69.1%)	228 (64.4%)	
Alcohol use status (%)					0.011
Former or current	424 (59.3%)	140 (52.4%)	63 (67.0%)	221 (62.4%)	
Never	291 (40.7%)	127 (47.6%)	31 (33.0%)	133 (37.6%)	
Smoking status (%)					0.004
Former or current	219 (30.6%)	62 (23.2%)	32 (34.0%)	125 (35.3%)	
Never	496 (69.4%)	205 (76.8%)	62 (66.0%)	229 (64.7%)	

^1^ Statistical significance among MDD, BP I, and BP II patient groups. *p*-value of age was measured by ANOVA, and other characteristics’ *p*-values were measured by Pearson’s chi-squared test.

**Table 2 medicina-59-00115-t002:** Results of the Cronbach’s alpha and McDonald’s omega reliability analyses.

Temperament	α	$\Omega_{t}$
Cyclothymic	0.85	0.87
Depressive	0.78	0.83
Irritable	0.79	0.82
Hyperthymic	0.74	0.79
Anxious	0.65	0.66
Total	0.88	0.9

**Table 3 medicina-59-00115-t003:** Factor loadings for 5 temperaments using exploratory factor analysis (varimax rotation).

	Factor 1	Factor 2
Cyclothymic	0.69	0.16
Depressive	0.76	−0.17
Irritable	0.65	0.25
Hyperthymic	0.04	0.87
Anxious	0.42	−0.00
Explained variance (%)	33.0	17.6

Factor loadings higher than 0.4 were grouped. Factor 1 (cyclothymic, depressive, irritable, and anxious) and Factor 2 (hyperthymic).

**Table 4 medicina-59-00115-t004:** Spearman’s correlation results of 5 temperaments in the TEMPS-A (short version).

	Cyclothymic	Depressive	Irritable	Hyperthymic	Anxious
Cyclothymic	1	0.49 ***	0.50 ***	0.18 ***	0.31 ***
Depressive		1	0.44 ***	−0.12 **	0.32 ***
Irritable			1	0.25 ***	0.23 ***
Hyperthymic				1	0.01
Anxious					1

Correlation significance ** *p* < 0.01, *** *p* < 0.001.

**Table 5 medicina-59-00115-t005:** Partial Spearman’s correlation between 5 temperaments and age.

Partial Correlation	Cyclothymic	Depressive	Irritable	Hyperthymic	Anxious
Age	−0.26 ***	−0.27 ***	−0.15 ***	−0.08 *	−0.00

Correlation significance * *p* < 0.05, *** *p* < 0.001. Gender, major diagnosis, education, employment, marital status, psychiatric first-degree family history, alcohol use status, and smoking status were controlled.

## Data Availability

The data presented in this study are available within the article and the Appendix A.

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
