# Peer review of "Korean Validation of the Short Version of the TEMPS-A (Temperament Evaluation of Memphis, Pisa, Paris, and San Diego Autoquestionnaire) in Patients with Mood Disorders"

_medicina, 2023, doi:10.3390/medicina59010115_

Round 1

Reviewer 1 Report

The article presents the verification of the short version of Korean TEMPS-A.

In this study, as emphasized by the authors, the short version of Korean TEMPS-A performed well in terms of validity and reliability among the clinical population - among patients with mood disorders. In general, Cronbach's alpha coefficients were significant. However, Cronbach's alpha coefficient for the "anxious temperament" subscale was 0.65, (0.66 - McDonald's omega coefficients). Although similar results have been obtained in previous studies in other countries, it is hardly satisfactory. Perhaps it would be advisable to refine this subscale even in the Korean population.

An exploratory factor analysis found two super-factors among the five temperaments and (generally) confirmed the research results in other populations.

The authors conducted many comparisons of the results obtained in the Korean studies with the results in other populations in relation to selected variables, including the gender of the subjects and the diagnosis of BD I, BD II, and MDD. It is interesting e.g. the result indicating no significant differences in the severity of "Irritable temperament" related to gender and the diagnosis of BD I, BD II, and MDD. Also interesting is the result indicating no differences in the intensity of the four temperaments between the sexes - with the exception of the cyclothymic scale/subscale. This result has been satisfactorily explained. This explanation confirms the usefulness of the tool in clinical trials, including comparative studies.

The authors, seeking to explain the discrepancies in Korean studies compared to studies in other countries/regions of the world, pointed to probable causes, including culture and the issue of representativeness of the study sample. I wonder if paying more attention to age (age range in the sample group: from 16 to 71 years old) might help eliminate or understand some of these differences. 

The authors mentioned the modification of the tool - but it is difficult to find information on this subject in the article.

To sum up, the analyzes presented in the article prove many advantages of the tool verified in Korean research. 

Author Response

The article presents the verification of the short version of Korean TEMPS-A.

In this study, as emphasized by the authors, the short version of Korean TEMPS-A performed well in terms of validity and reliability among the clinical population - among patients with mood disorders. In general, Cronbach's alpha coefficients were significant. However, Cronbach's alpha coefficient for the "anxious temperament" subscale was 0.65, (0.66 - McDonald's omega coefficients). Although similar results have been obtained in previous studies in other countries, it is hardly satisfactory. Perhaps it would be advisable to refine this subscale even in the Korean population.

>> Thank you for your constructive advice. Even though the low reliability coefficient may be due to the small number of items in the anxious temperament subscale (only 3 items), we agree that there is a need to refine the subscale to strengthen the internal consistency. We added this as a limitation to our discussion.

[Discussion] Page 8, lines 197-204

Similar to our study, some TEMPS-A-SV studies of other languages have also reported coefficients that are lower than 0.7 (Lin et al., 2018; Morvan et al., 2011; Nakato et al., 2016; Preti et al., 2013). In addition, as Cronbach’s alpha and McDonald’s omega coefficients are influenced by the number of items within a scale (Cortina, 1993; Viladrich et al., 2017), α = 0.65 and ω = 0.66 are acceptable values, considering the small number of items evaluating anxious temperament (3 items). Nevertheless, the reliability coefficients of anxious temperament subscale imply that there is a need to refine this subscale to strengthen its internal consistency.
An exploratory factor analysis found two super-factors among the five temperaments and (generally) confirmed the research results in other populations.
>> Thank you for your comment. We emphasized that our result was similar to the results of other validation studies.

[Discussion] Page 8, lines 210-211

This result was similar to the results of other validation studies conducted in different languages (Ristić-Ignjatović et al., 2014; Woodruff et al., 2011), which also found a two-factor structure.
The authors conducted many comparisons of the results obtained in the Korean studies with the results in other populations in relation to selected variables, including the gender of the subjects and the diagnosis of BD I, BD II, and MDD. It is interesting e.g. the result indicating no significant differences in the severity of "Irritable temperament" related to gender and the diagnosis of BD I, BD II, and MDD. Also interesting is the result indicating no differences in the intensity of the four temperaments between the sexes - with the exception of the cyclothymic scale/subscale. This result has been satisfactorily explained. This explanation confirms the usefulness of the tool in clinical trials, including comparative studies.

>> We appreciate the positive opinion of this reviewer. As you have said, we believe the TEMPS-A-SV is an effective tool in measuring affective temperaments and will be useful in clinical trials and comparative studies, as well as being helpful in diagnosis and treatment of mood disorders in the Korean clinical population.

The authors, seeking to explain the discrepancies in Korean studies compared to studies in other countries/regions of the world, pointed to probable causes, including culture and the issue of representativeness of the study sample. I wonder if paying more attention to age (age range in the sample group: from 16 to 71 years old) might help eliminate or understand some of these differences.

>> We appreciate the reviewer’s suggestion. We have added the age range as one of the probable causes of the correlation differences.

[Discussion] Page 9, lines 240-246

Also, this difference in correlations may be due to our sample group’s age range. The correlations of temperaments with age tend to be more significant when the standard deviation of age in the sample group is bigger (Hinić et al., 2013; Karam et al., 2005; Rózsa et al., 2008). Since the standard deviation of age in our sample is quite big (SD = 12.4), covering a larger age range, thus our analyses were more likely to identify correlations between temperaments and age.

The authors mentioned the modification of the tool - but it is difficult to find information on this subject in the article. To sum up, the analyzes presented in the article prove many advantages of the tool verified in Korean research.

>> Thank you for your question. Modification was mentioned to explain the process of trimming sentences to improve readability for Korean participants; there were no deletion or addition of items within the existing TEMPS-A-SV questionnaire. To prevent confusion, this sentence was edited to clarify the procedure.

[Measurement and procedures] Page 3, lines 100-101

After the review, the questionnaire was trimmed appropriately to improve legibility and fluency.

Reviewer 2 Report

a) I understand scores are not mutually exclusives, so the temperament is evaluated in a five-fold way along the scales, and not as "which one" each person displays as prevalent/dominant. If not, authors should also mention the prevalence of the five dominant temperaments and the distribution in diagnostic groups. 

I also understand the version is not a self-report one (TEMPS A's have different versions, self-evaluation and clinician's evaluation). That should be specified.

Author Response

  1. a) I understand scores are not mutually exclusives, so the temperament is evaluated in a five-fold way along the scales, and not as "which one" each person displays as prevalent/dominant. If not, authors should also mention the prevalence of the five dominant temperaments and the distribution in diagnostic groups.

>> Thank you for this comment. You are correct in understanding that the temperament scores are not mutually exclusive. We agree with the reviewer that the TEMPS-A-SV is not used to determine the “dominant” temperament, and each affective temperament is assessed separately, not in a mutually exclusive way. Therefore, we do not present a distribution or prevalence of the dominant temperament in diagnostic groups.

I also understand the version is not a self-report one (TEMPS A's have different versions, self-evaluation and clinician's evaluation). That should be specified.

>> Thank you for your comment. Our study validated the short version of TEMPS-A (self-report version), which is different from TEMPS-I (clinician evaluation version). We added the term ‘self-report’ in the introduction and methods to prevent confusion.

[Introduction] Page 2, line 48

It is a self-report questionnaire comprising 110 items.

[Measurement and procedures] Page 3, lines 96-97

TEMPS-A-SV is a self-report scale, and each item received a dichotomous score (Yes = 1 and No = 0).